# Multiple Porcine Innate Immune Signaling Pathways Are Involved in the Anti-PEDV Response

**DOI:** 10.3390/v15081629

**Published:** 2023-07-26

**Authors:** Youwen Zhang, Yulin Xu, Sen Jiang, Shaohua Sun, Jiajia Zhang, Jia Luo, Qi Cao, Wanglong Zheng, François Meurens, Nanhua Chen, Jianzhong Zhu

**Affiliations:** 1College of Veterinary Medicine, Yangzhou University, Yangzhou 225009, Chinadx120200161@stu.yzu.edu.cn (S.J.); hnchen@yzu.edu.cn (N.C.); 2Joint International Research Laboratory of Agriculture and Agri-Product Safety, Yangzhou 225009, China; 3Comparative Medicine Research Institute, Yangzhou University, Yangzhou 225009, China; 4Jiangsu Co-Innovation Center for Prevention and Control of Important Animal Infectious Diseases and Zoonoses, Yangzhou 225009, China; 5Swine and Poultry Infectious Diseases Research Center, Faculty of Veterinary Medicine, University of Montreal, St. Hyacinthe, QC J2S 2M2, Canada; francois.meurens@umontreal.ca; 6Department of Veterinary Microbiology and Immunology, Western College of Veterinary Medicine, University of Saskatchewan, Saskatoon, SK S7N 5E2, Canada

**Keywords:** PEDV, porcine innate immunity, pattern recognition receptor (PRR), signaling adaptors, knockout, siRNA

## Abstract

Porcine epidemic diarrhea virus (PEDV) has caused great damage to the global pig industry. Innate immunity plays a significant role in resisting viral infection; however, the exact role of innate immunity in the anti-PEDV response has not been fully elucidated. In this study, we observed that various porcine innate immune signaling adaptors are involved in anti-PEDV (AJ1102-like strain) activity in transfected Vero cells. Among these, TRIF and MAVS showed the strongest anti-PEDV activity. The endogenous TRIF, MAVS, and STING were selected for further examination of anti-PEDV activity. Agonist stimulation experiments showed that TRIF, MAVS, and STING signaling all have obvious anti-PEDV activity. The siRNA knockdown assay showed that TRIF, MAVS, and STING are also all involved in anti-PEDV response, and their remarkable effects on PEDV replication were confirmed in TRIF^−/−^, MAVS^−/−^ and STING^−/−^ Vero cells via the CRISPR approach. For further verification, the anti-PEDV activity of TRIF, MAVS, and STING could be reproduced in porcine IPEC-DQ cells treated with siRNAs. In summary, this study reveals that multiple pattern-recognition receptor (PRR) signaling pathways of porcine innate immunity play an important role in the anti-PEDV infection, providing new and useful antiviral knowledge for prevention and control of PEDV spreading.

## 1. Introduction

Porcine epidemic diarrhea virus (PEDV) is an intestinal coronavirus that targets the small intestinal epithelial cells of pigs. It causes damage to the small intestinal epithelial tissue, inducing intestinal congestion, swelling, and watery diarrhea in pigs, which leads to anorexia and body wasting in fattening pigs and high mortality rate in suckling piglets [1]. In recent years, with the emergence of new PEDV variant strains, PED broke out again globally, especially in Asian countries, causing great economic loss [2,3].

As a porcine enteric virus, PEDV belongs to the α coronavirus class and is a positive-sense single-stranded RNA virus [4]. The total length of the virus genome is 28 kb and it contains open reading frame (ORF)1a, ORF1b, spike protein (S), accessory protein ORF3, envelope protein (E), membrane protein (M) and nucleocapsid protein (N) genes [4]. The S protein encoded by the S gene consists of two domains, S1 and S2, with the interaction between S1 and host cell receptors as the first step of infection and the main determinant of viral anisotropy [5]. Additionally, PEDV encodes two large polyproteins, pp1a and pp1ab. Via the protease activity of non-structural proteins (Nsp) 3 and Nsp5, these two large polyproteins are further processed into 16 non-structural proteins, Nsp1 to Nsp16 [6]. In addition to its genetic diversity, PEDV has also evolved a variety of strategies to antagonize host innate antiviral defense for successful infection [7].

The innate immune system is known to be the first line of defense against infection caused by pathogens [8]. Pathogenic microorganisms produce conserved molecules called pathogen-associated molecular patterns (PAMPs) when they invade the organism or replicate in the host organism [8]. Host cells express receptors for these molecules, named pattern recognition receptors (PRRs), which are composed of toll-like receptors (TLRs), RIG-I-like receptors (RLRs), NOD-like receptors (NLRs), C-type lectin receptors (CLRs), and cytosolic DNA receptors (CDRs) [9]. These receptors recognize PAMPs and activate intracellular cascade signaling pathways comprising adaptors, kinases, and transcription factors, ultimately initiating intracellular gene transcription or protease cleavage of cytokine precursors [9]. Thus, the produced antiviral interferons (IFNs), inflammatory factors, and chemokines play an anti-infection role while profoundly influencing subsequent adaptive immune responses [9,10].

The mutual evolution of virus and innate immune response of the host leads to significant viral diversification and enhanced host antiviral response [11]. In the competition between viruses and host cells, many viruses, including coronaviruses, have evolved various strategies to evade or disrupt antiviral immunity, such as type I and type III interferons (IFNs) [12]. There is increasing evidence that several PEDV proteins such as nsp1, nsp3, nsp5, nsp8, nsp14, nsp15, nsp16, E, M, and N can resist host IFN signaling [13]. In addition, PEDV has been reported to inhibit IFN induction via different mechanisms [13,14]. As for innate immunity for recognition of and defense against PEDV, it has not been totally understood until now. In this study, we demonstrated that PEDV can activate multiple PRR-mediated signaling pathways to induce anti-PEDV activity. This study provides new insight into host innate immune responses against PEDV infection.

## 2. Materials and Methods

### 2.1. Cells, Virus and Reagents

PEDV-permissive Vero cells and human embryonic kidney (HEK) 293T cells were cultured in Dulbecco’s Modified Eagle’s Medium (DMEM; Thermo Fisher Scientific, Waltham, MA, USA) supplemented with 10% fetal bovine serum (FBS, Thermo Fisher Scientific) and 100 U/mL of penicillin plus 100 g/mL streptomycin. IPEC-DQ cells were cultured in Roswell Park Memorial Institute (RPMI) medium (Hyclone Laboratories, Logan, UT, USA) containing 10% FBS with penicillin/streptomycin. All cells were maintained at 37 °C with 5% CO_2_ in a humidified incubator. The PEDV strain used in this study was a mutant strain of PEDV AJ1102 (GenBank: JX188454.1). Poly(I:C)-LMW, poly(I:C)-HMW, 2′3′-cGAMP and poly(dA:dT) were all obtained from InvivoGen (Hongkong, China). Mouse anti-glyceraldehyde-3-phosphate dehydrogenase (GAPDH) mAb, mouse anti-green fluorescence protein (GFP) mAb, and mouse anti-β-Actin mAb were all brought from TransGen (Beijing, China). Mouse anti-MAVS mAb was from Santa Cruz Biotechnology (Dallas, TX, USA), rabbit anti-TRIF pAb was from Biodargon (Beijing, China), and rabbit anti-STING pAb was bought from Proteintech (Wuhan, China).

### 2.2. Preparation of Monoclonal Antibody against N Protein of PEDV

The PEDV N gene amplified by PCR from PEDV cDNA using the primers shown in Table 1 was inserted into the *EcoR* V and *Sal* I sites of pENTR4-2HA vector. The resultant recombinant pENTR4 vector was subjected for LR recombination with pDEST527 (Thermo Fisher Scientific) to construct the pDEST527-N prokaryotic expression vector. After induction at 25 °C for 12 h with 1 mM IPTG, the soluble N protein was expressed in transformed *E. coli* (DE3/BL21). The expressed N protein was purified via nickel column according to the instructions of the His-tag Protein Purification Kit (Beyotime, China). Female 6-week-old BALB/c mice were immunized four times with 30 μg purified N protein plus 1:1 immune adjuvant (QuickAntique-Mouse5w, Biodragon, China) each, via multiple subcutaneous injections in the back. The monoclonal antibody (mAb) of N protein was obtained via regular hybridoma technology from the immunized mice. The reaction specificity of N mAb is shown as in Appendix A.

### 2.3. Quantitative Reverse Transcription Polymerase Chain Reaction (RT-qPCR)

The total RNA of the Vero cells after PEDV infection or agonist stimulation was extracted with TRIzol reagent (Thermo Fisher Scientific). The extracted RNA was reverse-transcribed into cDNA with the HiScript^®^1st Strand cDNA Synthesis Kit (Vazyme, China). qPCR was performed in 20 μL reactions with the Cham Q Universal SYBR qPCR Master Mix (Vazyme, China) using the StepOne Plus real-time PCR system (Applied Biosystems, Foster City, CA, USA). The qPCR primers were as shown in Table 2. Results of the relative mRNA expression were calculated via the 2^−ΔΔCt^ method, with β-Actin used as the internal reference control.

### 2.4. Western Blot Analysis

The treated cells were collected and lyzed in a radio-immunoprecipitation assay (RIPA) buffer (50 mM Tris pH 7.2, 150 mM NaCl, 1% sodium deoxycholate, 1% Triton X-100). The extracted whole-cell protein was separated via sodium dodecyl sulfate-polyacrylamide gel electrophoresis (SDS-PAGE), and then the protein in gel was transferred to a polyvinylidene fluoride (PVDF) membrane. The membrane was blocked with Tris-buffered saline with Tween (TBST) solution containing 5% skim milk at room temperature for 1 h and incubated overnight at 4 °C with mouse PEDV N mAb (1:1000) as specific primary antibody or other primary antibodies. Next, the membrane was washed with TBST 3 times each for 3–5 min, and then incubated with HRP-labeled goat anti-mouse or rabbit IgG (1:10,000, Transgen Biotech, Beijing, China) as the secondary antibody. The protein signals were visualized using an imaging system (Tanon, China) with the ECL chemiluminescence detection system (Tanon, China) according to the manufacturer’s instructions.

### 2.5. Virus TCID50 Titration

Vero cells cultured in 96-well plates were infected with PEDV in 10-fold continuous dilutions at 37 °C for 2 h, and the supernatants were replaced with fresh DMEM containing 2% FBS. After 4 to 5 days of infection, Vero cells showed significant cytopathic effect (CPE), manifested as cell shedding. The virus titer was calculated according to the Reed–Muench method and expressed as 50% tissue culture infection dose (TCID50).

### 2.6. Plaque Assay

A suitable number of Vero cells were seeded into the different cell plates (1.5 × 10^6^ cells/well in 6-well plate and 3–4 × 10^5^ cells/well in 24-well plate). The PEDV sample to be tested was diluted 10^−1^–10^−4^ times to infect cells in the plate wells. After 2 h of viral infection, the monolayer cells were immediately and slowly covered with 1:1 mixed 1.6–2% low-melting agarose solution with DMEM medium containing 4% FBS. About 3–5 days after infection, cells were fixed and stained with crystal violet dye containing 4% polyformaldehyde. After staining, the cells were rinsed gently and directly with tap water until the clear plaques appeared, then photographed.

### 2.7. RNA Interference by siRNA

All siRNAs targeting monkey and porcine TRIF, MAVS, and STING were designed and synthesized by Thermo Fisher Scientific. The knockdown efficiency of individual siRNAs for monkey cells has already been tested in our previous work [15]. To determine the knockdown efficiency of designed porcine siRNA sequences (Table 3). IPEC-DQ cells were transfected with 50–150 nM individual siRNAs with Lipofectamine 2000 (Thermo Fisher Scientific), and the expression of corresponding proteins were measured via Western blotting using anti-TRIF antibody (1:1000), anti-MAVS antibody (1:1000), and anti-STING antibody (1:2000), respectively. For the validated siRNA duplexes, Vero cells or IPEC-DQ cells in 12-well plates with 70–80% monolayers were transfected with 100 nM siRNAs using Lipofectamine 2000 and used for subsequent experiments.

### 2.8. Preparation of TRIF^−/−^, MAVS^−/−^ and STING^−/−^ Vero Cells by CRISPR/Cas9 Approach

The clustered, regularly interspaced short palindromic repeat guide RNAs (CRISPR gRNAs) were designed based on the first exons of monkey STING (GenBank: CM014354.1), TRIF (GenBank: CM014354.1), and MAVS (GenBank: CM014354.1). For each gene, 2–3 gRNAs were selected (Table 4), and the annealed gRNA encoding DNA pairs were ligated with the *Bbs*I digested vector pSpCas9(BB)-2A-GFP (pX458, Addgene, Watertown, NY, USA). Subsequently, each gRNA expressing pX458 was transfected into Vero cells using Lipofectamine 2000, and the GFP-positive cells were sorted from transfected cells using a BD FACSAria III Sorter. The individual Vero cell clones obtained via limited dilution from the sorted GFP expressing cells were screened via PCR using the designed primers (Table 4). The PCR products were cloned into T vector, using the pClone007 Versatile Simple Vector Kit (TsingKe Biological Technology, Beijing, China). The inserted fragments were multiply sequenced, and the sequences were analyzed for base insertion and deletion (indel) mutations, based on which two TRIF^−/−^, three MAVS^−/−^ and four STING^−/−^ Vero cell clones were obtained (Appendix A).

### 2.9. Statistical Analysis

The software program Image J 5.0 was used to measure the gray values of Western blot protein bands. The data were represented by mean ± SD and analyzed using GraphPad Prism 7.0. The *t* test was used to determine whether there was any significant difference between the results: *p* < 0.05 (*) and *p* < 0.01 (**). *p* < 0.05 indicated statistical significance.

## 3. Results

### 3.1. Cell Infection and Titer Determination of PEDV Strain

The PEDV strain used in the study was able to effectively infect Vero cells and cause significant cytopathic effect (CPE). Both TCID50 and plaque assays were performed in Vero cells to determine the virus titer. As shown in Figure 1A,B and Appendix A, the titers measured via TCID50 assay and plaque assay were 10^4.75^/0.1 mL and 4 × 10^5^ PFU/0.1 mL at 72 h post-infection, respectively.

### 3.2. Detection of Activated IFN Levels and Transfected Adaptor Proteins in Vero Cells

Some studies have reported that an interferon (IFN) defect exists in Vero cells [16], whereas others have implied that Vero cells were able to produce IFN [17]. In this article, the anti-PEDV activity mediated by innate immune signaling adaptor proteins was studied mainly in Vero cells, and IFNs are an important effector affecting the antiviral results. To overcome this potential issue, we treated Vero cells with RLRs RIG-I/MDA5 agonist poly (I:C)-LMW and TLR3 agonist poly (I:C)-HMW for 12 h and 36 h, respectively, and then the collected cells were analyzed for IFN and IFN-stimulated gene (ISG) expressions via RT-qPCR. As shown in Figure 2A, the expression levels of IFN-α, IFN-β, and ISG15 genes were significantly upregulated upon both stimulations, indicating that the downstream IFN and ISG genes could be normally expressed in our Vero cells.

We previously cloned and characterized the nine porcine innate immune signaling adaptors which represent and reflect all currently known innate immune signaling pathways [18,19]. The recombinant pEGFP plasmids encoding MAVS, TRIF, STING, MyD88, RIPK2, ASC, CARD9, BCL10, MALT1, and control vector pEGFP-N1 were transferred into Vero cells for 24 h, and the expressions of transfected adaptor proteins were determined via Western blotting (WB). As shown in Figure 2B, all the adaptor proteins could be correctly expressed in the transfected Vero cells, although the expression level of TRIF was poor.

### 3.3. The Anti-PEDV Activities of Ectopic Porcine Signaling Adaptors in Vero Cells

To explore the impacts of nine adaptor proteins on PEDV replication, the adaptor-transfected Vero cells were infected with PEDV, and the PEDV replication was examined by measuring the viral N gene transcription by RT-qPCR and protein expression via Western blotting, respectively. The RT-qPCR results showed that all the signaling adaptors decreased the expressions of PEDV N mRNA at various levels at 48 h and 72 h post-infection, with TRIF, MAVS, and STING being very significant in inhibition of PEDV at 72 h post-infection (Figure 3A). The Western blotting results showed that TRIF, MAVS, and STING were most effective in inhibition, while MyD88, RIPK2, ASC, and CARD9-BCL10-MALT1 (CBM) were less effective in inhibiting N protein expression at 72 h post-infection (Figure 3B). The collected cell supernatants were measured for PEDV titer via plaque assay, which further confirmed the above results (Figure 3C and Appendix A).

### 3.4. The Antiviral Activities of Endogenous TRIF, MAVS, and STING Signaling against PEDV Replication

Based on the above screening results and considering the significance of the nucleic acid-sensing signal in the antiviral activity, TRIF, MAVS, and STING were chosen and their endogenous signaling was investigated for antiviral roles in PEDV infection. Several agonists were used to trigger endogenous innate signaling, including poly(I:C)-HMW for TLR3-TRIF signaling, poly(I:C)-LMW for RIG-I/MDA5-MAVS signaling, 2′3′-cGAMP for STING signaling, and poly(dA:dT) for cGAS-STING signaling. RT-qPCR (Figure 4A,B), WB detection (Figure 4C), and plaque assay (Figure 4D and Appendix A) all demonstrated that the poly(I:C)-HMW-induced TRIF signaling, poly(I:C)-LMV-induced MAVS signaling, 2′-3′-cGAMP, and poly(dA:dT)-activated STING signaling possess obvious anti-PEDV activity. The anti-PEDV effect in Vero cells was more obvious at 72 h post-infection than at 48 h post-infection (Figure 4A–C).

### 3.5. Knockdown of Endogenous TRIF, MAVS, and STING Enhances PEDV Replication

To further explore the roles of endogenous TRIF, MAVS, and STING in PEDV replication, the previous validated siRNAs were used to knock down TRIF, MAVS, and STING in Vero cells prior to PEDV infection [15]. In the TRIF, MAVS, and STING siRNA-treated Vero cells, PEDV N mRNA and N protein expressions were obviously increased at 72 h post-infection relative to control siRNA-treated cells (Figure 5A,B). Similarly, compared with control siRNA-treated cells, PEDV titers were obviously heightened in TRIF, MAVS, and STING siRNA-treated cells at 72 h post-infection (Figure 5C and Appendix A). These results suggest that endogenous TRIF, MAVS, and STING are all part of the host defense mechanism and necessary for inhibition of PEDV replication.

### 3.6. PEDV Replication Was Enhanced in TRIF^−/−^, MAVS^−/−^ and STING^−/−^ Vero Cells

In order to further verify the anti-PEDV infection effects of STING, TRIF, and MAVS in Vero cells, we used CRISPR/Cas9 gene editing technology to construct STING, TRIF, and MAVS knockout Vero cell lines, respectively. Clones of TRIF^−/−^, MAVS^−/−^, and STING^−/−^ Vero cells were obtained and used for PEDV infection together with normal Vero cells as a control. Compared with the control Vero cells (WT), the expression of N gene mRNA and N protein in TRIF^−/−^, MAVS^−/−^, and STING^−/−^ Vero cells after PEDV infection and the virus content in culture supernatant were significantly increased (Figure 6A–C and Appendix A). Additionally, WB again confirmed that the adaptor protein expression in the corresponding knockout cells disappeared as expected (Figure 6B). These results further proved the important roles of TRIF, MAVS, and STING in host defense against PEDV infection, especially the important influence of TRIF and MAVS in PEDV infections.

### 3.7. Knockdown of Endogenous TRIF, MAVS, and STING in IPEC-DQ Cells Promoted PEDV Replication

PEDV is a diarrhea virus prevalent in pigs. So as to further detect the effects of endogenous TRIF, MAVS, and STING on the replication of PEDV, we used relevant IPEC-DQ cell line of piglet jejunal epithelium for PEDV infection. First, different individual siRNA sequences of porcine TRIF, MAVS, and STING were transfected into IPEC-DQ cells at a concentration of 50–150 nM, respectively, and cells were collected 24 h later for detection to determine siRNA knockdown efficiency. In accordance with the knockdown effect on protein level detected via WB (Figure 7A–C), STING 11 siRNA, MAVS 2329 siRNA, and TRIF 5800 siRNA were finally selected for subsequent experiments at a working concentration of 100 nM.

The three siRNA duplexes and negative control siRNA duplex were transfected into IPEC-DQ cells using Lipofectamine 2000 for 24 h, followed by PEDV infection. At 72 h post-infection, RT-qPCR, WB, and plaque assay showed that when TRIF, MAVS, and STING in IPEC cells were silenced by corresponding siRNA, N gene transcription, N protein expression, and virus replication levels of PEDV were slightly increased compared with the control group (Figure 7D–F and Appendix A). These results further indicated that a variety of porcine innate immune signaling pathways exert the anti-PEDV function.

## 4. Discussion

Porcine epidemic diarrhea (PED) is a swine disease with worldwide distribution which has important economic significance, posing a huge threat to the swine industry. At present, PEDV variants have emerged and evolved persistently [20]. Although there are several PED vaccines available on the market, protective immunity from vaccination depends on both effective innate immunity and adaptive immunity [21]. Innate immune response is the first line of host defense during infection and plays a crucial role in early recognition of and immune protection against virus infection. However, PEDV does not elicit a robust antiviral IFN response; the interaction between PEDV and host immunity is complex, and its mechanism has not been fully understood until recently [22]. To understand and maximize the potential of PEDV innate immunity, we examined the anti-PEDV innate immune response globally and completely by investigating the signaling adaptors which represent and reflect all currently known innate immune signaling pathways. Our results and findings indicated that multiple innate immune signaling pathways are involved in the recognition of and defense against PEDV infection.

Among different PRR families, three types of PRRs have been identified in the recognition of viral nucleic acids, including RLR detection of viral RNA in the cytoplasm [23], TLR recognition of viral RNA or DNA in the endosome [24], and CDR detection of viral DNA in the cytoplasm [25]. Here, we found and confirmed the significant roles of the RLR adaptor MAVS, the TLR3 adaptor TRIF, and the CDR adaptor STING in anti-PEDV response. Additionally, other innate signaling adaptors MyD88, RIPK2, ASC, and CARD9-BCL10-MALT1 (CBM complex) also appeared to play a role in anti-PEDV response. MyD88 is the common signaling adaptor for all TLRs except TLR3 [26]. RIPK2 is the signaling adaptor for NLRs NOD1 and NOD2, whereas ASC is the adaptor for other NLRs [27]. The signaling adaptor complex of CBM is for all CLRs [28]. Among these various PRRs, the roles of some PRRs such as NLRs and CLRs have never been investigated and need to be validated and investigated during PEDV infection.

Pigs at all life stages can be infected with PEDV; however, both the morbidity and mortality of suckling piglets under 7 days of age are very high. The mortality of fattening pigs and sows after infection is very low, and most infected pigs show recessive or transient diarrhea. Adult pigs are raised with mucosal immunity and often immunized with vaccines compared with newborn piglets. It is very likely that there is a difference in the PRR-adaptor cell signaling pathways between the intestinal epithelial cells of young vs. older pigs. Additionally, other resistance mechanisms may also be at play, which differs between young vs. old pigs.

We previously showed that multiple innate immune signaling pathways are involved in the recognition of and defense against porcine reproductive and respiratory syndrome virus (PRRSV), a member of *Arteriviridae* family [15]. Both PRRSV and PEDV harbor positive-sense single-stranded genomic RNA, which may confer similarities in the induced innate immune responses. Furthermore, the phenotype may be general to other viruses, that is, many virus infections are recognized and counteracted by several innate immune PRR-mediated signaling pathways. Subsequently, several intriguing questions arise: (1) What is the relative contribution of each PRR signaling pathway in the anti-PEDV innate immune response? (2) What is the relationship between different PRR signaling pathways in PEDV infection? (3) How does PEDV evade each of these PRR signaling pathways? Answering these questions will reveal an intricate and delicate interaction network between PEDV and host, which deserves further researches in the future.

In summary, our results suggest that multiple porcine PRR-mediated signaling pathways are involved in PEDV recognition and defense. These results deepen our understanding of PEDV innate immunity and help maximize the potential of innate immunity to control PEDV infection.

## Figures and Tables

**Figure 1 viruses-15-01629-f001:**
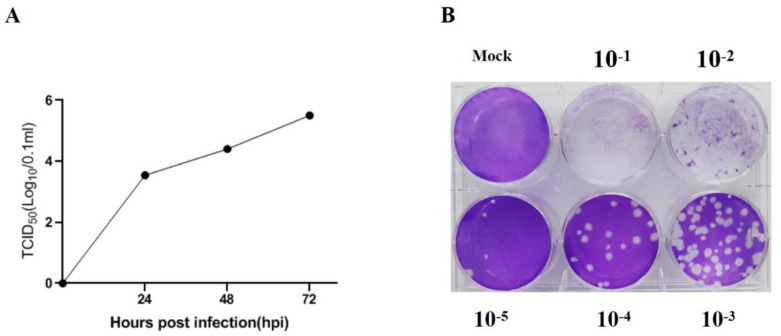
Titration of PEDV via TCID50 assay (**A**) and plaque assay (**B**), respectively.

**Figure 2 viruses-15-01629-f002:**
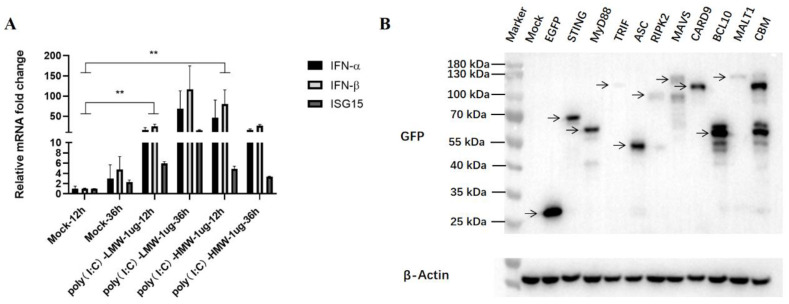
Detection of IFN levels and the adaptor expressions in Vero cells. (**A**) RT-qPCR detection of interferons (IFNs) and related genes in Vero cells. ** *p* < 0.01. (**B**) Expressions of 9 signaling adaptor proteins in transfected Vero cells, which are fused with GFP and marked with arrows.

**Figure 3 viruses-15-01629-f003:**
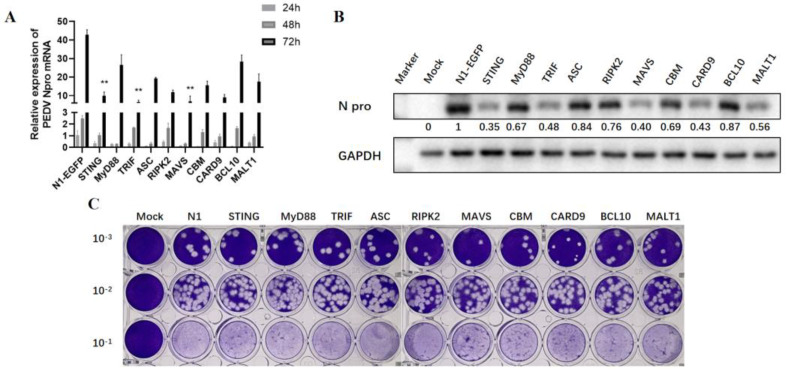
Detection of the anti-PEDV activity of exogenous adaptor proteins in Vero cells. (**A**) The effect of 9 adaptor proteins on PEDV N gene transcription at different time points was detected via RT-qPCR. ** *p* < 0.01. (**B**) The effect of 9 adaptor proteins on N protein expression of PEDV at 72 h post-infection was detected via WB. The gray values of each N protein bands after normalized by GAPDH are shown below the corresponding protein bands. (**C**) The effect of 9 adaptor proteins on virus amounts in cell supernatant after 72 h infection was detected via plaque assay.

**Figure 4 viruses-15-01629-f004:**
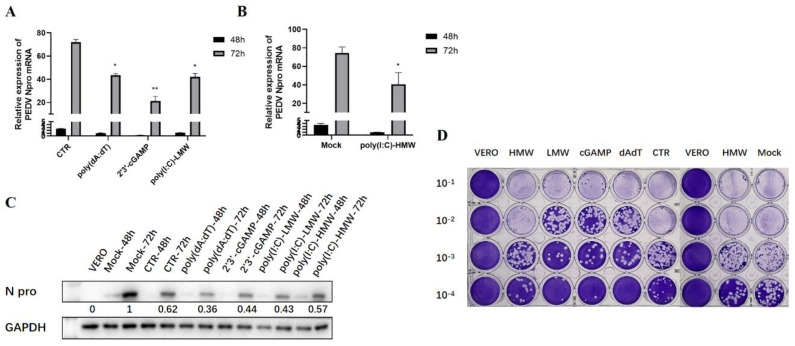
The effects of endogenous TRIF, MAVS, and STING signaling on PEDV replication in Vero cells. (**A**,**B**) The effect of different stimulants on PEDV replication at both 48 h and 72 h post-infection was detected via RT-qPCR. CTR denotes the transfection control and Mock the non-treatment control. * *p* < 0.05, ** *p* < 0.01. (**C**) The effect of stimulation on PEDV replication at both 48 h and 72 h post-infection was detected via WB. The gray values of each N protein bands after normalization by GAPDH are shown below the corresponding protein bands. (**D**) The effect of stimulants on the amount of virus in the supernatant after 72 h infection was detected via plaque assay.

**Figure 5 viruses-15-01629-f005:**
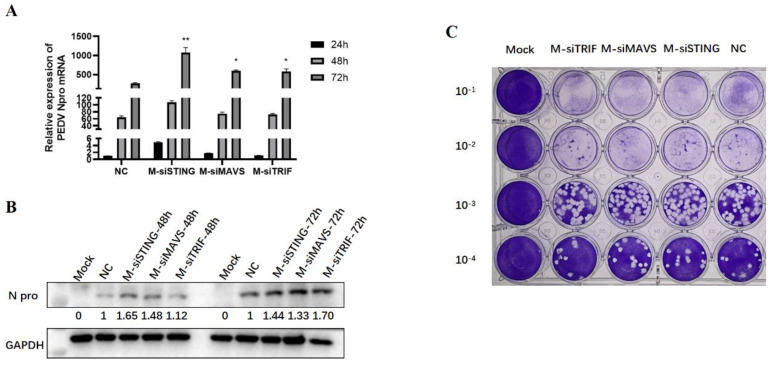
The effects of knockdown of endogenous TRIF, MAVS, and STING on PEDV replication in Vero cells. (**A**) The effect of signaling adaptor knockdown on PEDV N gene expression was detected via RT-qPCR. NC denotes the negative control siRNA. * *p* < 0.05, ** *p* < 0.01. (**B**) The effect of signaling adaptor knockdown on PEDV N protein expression. The gray values of each N protein bands after normalized by GAPDH are shown below the corresponding protein bands. (**C**) The effect of signaling adaptor knockdown on PEDV replication after 72 h infection in Vero cell was detected via plaque assay.

**Figure 6 viruses-15-01629-f006:**
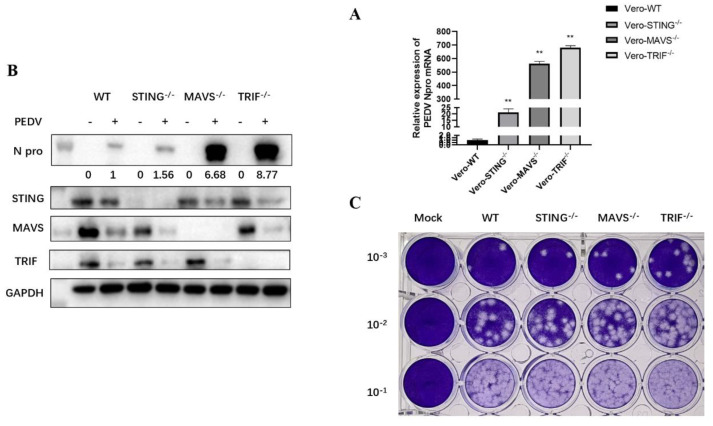
PEDV replication was increased in TRIF^−/−^, MAVS^−/−^, and STING^−/−^ Vero cells. (**A**) RT-qPCR was used to detect PEDV N gene mRNA expression in knockout (KO) cells and wild-type (WT) normal Vero cells. ** *p* < 0.01. (**B**) The expression of PEDV N protein in knockout cells was detected via WB. The gray values of each N protein bands after normalization by GAPDH are shown below the corresponding protein bands. (**C**) The PEDV content in the knockout cell supernatants was detected via plaque assay.

**Figure 7 viruses-15-01629-f007:**
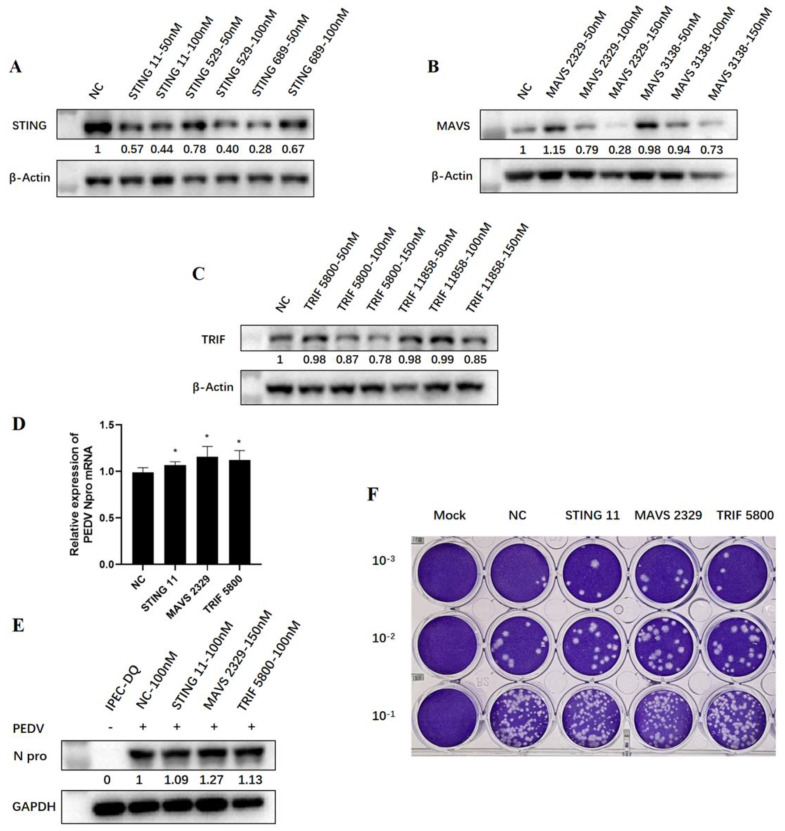
Effects of siRNA knockdown of endogenous TRIF, MAVS, and STING on PEDV replication in IPEC-DQ cells. (**A**–**C**) WB detection of TRIF, MAVS, and STING siRNA knockdown efficiency by different siRNAs. The gray values of each adaptor protein bands after normalization by β-Actin are shown below the corresponding protein bands. (**D**) The effect of signaling adaptor knockdown on PEDV replication was detected via RT-qPCR. * *p* < 0.05. (**E**) The impact of signaling adaptor knockdown on PEDV replication was detected via WB. The gray values of each N protein bands after normalization by GAPDH are shown below the corresponding protein bands. (**F**) The effect of adaptor knockdown on the level of supernatant PEDV at 72 h post-infection was detected via plaque assay.

**Table 1 viruses-15-01629-t001:** PCR primers used for PEDV N gene amplification.

Prime Names	Sequences (5′-3′)
PEDV-N-F	GGGCCGTCGACATGGCTTCTGTCAGTTTTCAGGATCG
PEDV-N-R	GGGCCGATATCATTTCCTGTATCGAAGATCTCGTTGATAATTTCAAC

Note: the restriction enzyme sites are underlined.

**Table 2 viruses-15-01629-t002:** The qPCR primers used for the detection of mRNA expression.

Prime Names	Sequences (5′-3′)
PEDV-N-f	CAAGAACAGAAACCAGTCAAATGACC
PEDV-N-R	AGAGTGGAGGAGAATTCCCAAGG
M-IFN-α-F	ATCTGCTCTCTGGGCTGTGATCT
M-IFN-α-R	TTCAGACAGGAGAAAGGAGAGATTCT
M-IFN-β-F	AAATTGCTCTCCTGTTGTGCTTCT
M-IFN-β-R	AAGCCTTCCATTCAATTGCCA
M-ISG15-F	CTCTGAGCATCCTGGTGAGGAA
M-ISG15-R	CGAAGGTCAGCCAGAACAGGT
M-β-Actin-F	AGAAGATGACCCAGATCATGTTTG
M-β-Actin-R	ATCCATCACGATGCCAGTGGTA

Note: M denotes monkey.

**Table 3 viruses-15-01629-t003:** The designed siRNA sequences for porcine TRIF, MAVS, and STING genes.

siRNA Names	Sequences (5′-3′)
P-STING-11-F/R	CCAGCCUGCAUCCAUCCAUTT AUGGAUGGAUGCAGGCUGGTT
P-STING-689-F/R	CCGACCGUGCUGGCAUCAATT UUGAUGCCAGCACGGUCGGTT
P-STING-529-F/R	GCUCGGAUCCAAGCUUAUATT UAUAAGCUUGGAUCCGAGCTT
P-MAVS2329-F/R	CCACCACAGAGAUCUUUAATT UUAAAGAUCUCUGUGGUGGTT
P-MAVS3138-F/R	GGCUGCACUACUGUAUUAUTT AUAAUACAGUAGUGCAGCCTT
P-TRIF5800-F/R	GCCUGUCCUUUACCCUUUATT UAAAGGGUAAACGACACCCTT
P-TRIF11858-F/R	GGGUUCAUCACAUUAAUAATT UUAUUAAUGUGAUGAACCCTT
siNC-F/R	UUCUCCGAACGUGUCACGUTT ACGUGACACGUUCGGAGAATT

Note: The P denotes porcine.

**Table 4 viruses-15-01629-t004:** The CRISPR gRNA encoding DNA sequences and PCR primers for monkey TRIF, MAVS, and STING genes.

Prime Names	Sequences (5′-3′)
M-STING gRNA1-F/R	CACCGTGGATGGATGCAGACTGGAG AAACCTCCAGTCTGCATCCATCCAC
M-STING gRNA2-F/R	CACCGCCATCCATCCCGTGTCCCAG AAACCTGGGACACGGGATGGATGGC
M-STING gRNA3-F/R	CACCGCTGGGACAGCTGTTAAATG AAACCATTTAACAGCTGTCCCAGC
M-TRIF gRNA1-F/R	CACCGTAGGCCACGTCCCGCAGCG AAACCGCTGCGGGACGTGGCCTAC
M-TRIF gRNA2-F/R	CACCGATGAGGCCCGAAACCGGTGT AAACACACCGGTTTCGGGCCTCATC
M-MAVS gRNA1-F/R	CACCGTCTTCAGTACCCTTCAGCGG AAACCCGCTGAAGGGTACTGAAGAC
M-MAVS gRNA2-F/R	CACCGCTGGTAGCTCTGGTAGACAC AAACGTGTCTACCAGAGCTACCAGC
M-STING-F/R	TCGCAGAGACAGGAGCTTTG GGCTGCAGACCCCATTTAAC
M-TRIF-F/R	ACTGAAGGCTGATGCAGCG TTTCCAAGTTGCTGGCCAGG
M-MAVS-F/R	GCTCTTCTGGCTTTCTTGGCG GCTCAGCCTGGATCTACACCC

Note: M denotes monkey.

## Data Availability

More data are available upon request.

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
