# Peer review of "Multiple Porcine Innate Immune Signaling Pathways Are Involved in the Anti-PEDV Response"

_viruses, 2023, doi:10.3390/v15081629_

Round 1
Reviewer 1 Report
Review paper Viruses 2023 “Multiple porcine innate immune signaling pathways are involved in the anti-PEDV response” of Zhang Y et al.
The authors have performed different type of in vitro studies to evaluate the innate immune response and in particular the role herein of a couple of specific innate signaling proteins such as TRIF, MAVS and STING adaptors following PEDV infection. Using KO cells and siRNA knockdown studies the individual contribution of these PRR-adaptor proteins have been demonstrated on PEDV replication in Vero cells and with siRNA studies in porcine IPEC cells.
Overall, manuscript is well written and scientifically sound, although some aspects should need to be revised as outlined below to make it eligible for publication.
Major items:
-M&M: the creation of KO cell line clones using Crispr-Cas technology is applied for the Vero cell line only, what is rational to not focus on the more relevant IPEC cell line for conducting this test and evaluate and strengthen the investigation of the role of these adaptors in a host-species cell line?
-Results Figure 3: what is the phenotype once focusing on another (properly benchmarked) PEDV-epitope like S-protein for which suitable mAbs exist that are widely used within scientific community? Current study focuses only on the use of a N-protein based mAb using the own-made hybridoma clone, which in addition is not properly explained in terms of QC steps. Do authors expect similar observations?
-Results Figure 7: the impact of adaptor protein silencing using siRNA in IPEC cells seem to be less significant as observed in the Vero cell line. Do the authors have an explanation and rational for this observation? What is the relevance of this observation towards translating the infectivity potential of PEDV in its natural host?
-Discussion/rationale: Although the basic objective of the study is to characterize the role of diverse innate immune pathways by adaptor proteins through applying invitro studies using Vero cells, it is known that predominantly young piglets are susceptible for infection via their intestinal epithelium, whereas older pigs are not anymore. It would be of interest to put the data in the correct context within the discussion; would there be a difference in the PRR (-adaptor protein) signaling pathways between intestinal epithelial cells of young vs older animals? Are other mechanisms at play, or is it only the (acquired) resistance which differs between young vs old?
-quantification of the plaque assays has not been performed / is not included in the manuscript. Without such quantification (plaque count), it is difficult for the reader to judge the scientific value of the observations. Have enough replicates been performed?
Minor items:
-Materials&Methods: have the monoclonal antibodies used for the adaptor proteins been verified to cross-detect porcine host-proteins?
-M&M: please include some (supplementary) data showing the N-specific mAb generated indeed exerts the target antigen-specificity and affinity as aimed for.
-Results figure 2b: what is the explanation for the other expression bands observed in some of the transfected cell lines outside the corresponding adaptor protein? Is it reflecting cross-reactivity of the Ab’s used?
-line 39: Belongs to the alpha-coronavirus family?
-line 201: despite the fact?
-Results line 211 typo: PRRSV = PEDV
-line 220 proteins (plural)
-Figure 3: positioning in of the A and B panels
Line 314: ‘worldwide swine disease’ : do not know if this is accepted phrasing / swine disease with worldwide distribution?
no comments / minor language review required
Author Response
Comments and Suggestions for Authors
Review paper Viruses 2023 “Multiple porcine innate immune signaling pathways are involved in the anti-PEDV response” of Zhang Y et al.
The authors have performed different type of in vitro studies to evaluate the innate immune response and in particular the role herein of a couple of specific innate signaling proteins such as TRIF, MAVS and STING adaptors following PEDV infection. Using KO cells and siRNA knockdown studies the individual contribution of these PRR-adaptor proteins have been demonstrated on PEDV replication in Vero cells and with siRNA studies in porcine IPEC cells.
Overall, manuscript is well written and scientifically sound, although some aspects should need to be revised as outlined below to make it eligible for publication.
Thanks for the appreciation by the reviewer.
Major items:
-M&M: the creation of KO cell line clones using Crispr-Cas technology is applied for the Vero cell line only, what is rational to not focus on the more relevant IPEC cell line for conducting this test and evaluate and strengthen the investigation of the role of these adaptors in a host-species cell line?
Answer: Compared with IPEC cells, PEDV used in this study replicates well on Vero cells, with obvious cell pathogenic effect (CPE), which is easy for follow-up study and observation. In contrast, there is little CPE in PEDV infected IPEC cells. In addition, our previous work has proved that the signaling pathway adaptor proteins in monkey cells Marc-145 cells have anti-porcine virus effect (Vaccines. 2021; 9(10): 1176). Therefore, the focus of the early experiments was placed on Vero cells.
-Results Figure 3: what is the phenotype once focusing on another (properly benchmarked) PEDV-epitope like S-protein for which suitable mAbs exist that are widely used within scientific community? Current study focuses only on the use of a N-protein based mAb using the own-made hybridoma clone, which in addition is not properly explained in terms of QC steps. Do authors expect similar observations?
Answer: In order to detect the PEDV replication specifically, we tried to prepare the monoclonal antibodies (mAb) against both N and S protein at the same time for the follow-up study at first. However, only the N mAb but not S mAb works in Western blotting detection of PEDV. The S mAb can be used for detection of PEDV in IFA, despite of weak signal, giving consistent results with those of N mAb detection in general.
-Results Figure 7: the impact of adaptor protein silencing using siRNA in IPEC cells seem to be less significant as observed in the Vero cell line. Do the authors have an explanation and rational for this observation? What is the relevance of this observation towards translating the infectivity potential of PEDV in its natural host?
Answer: This phenomenon may be due to two aspects: one is the less knockdown efficiency of siRNA in IPEC cells relative to that of Vero cells. Another is likely the greater redundancy between different adaptor signaling in the anti-PEDV response in IPEC cells relative to Vero cells. It suggests that in order to exert the best anti-PEDV response, combination of several adaptor signaling must be adopted in the natural host.
-Discussion/rationale: Although the basic objective of the study is to characterize the role of diverse innate immune pathways by adaptor proteins through applying in vitro studies using Vero cells, it is known that predominantly young piglets are susceptible for infection via their intestinal epithelium, whereas older pigs are not anymore. It would be of interest to put the data in the correct context within the discussion; would there be a difference in the PRR (-adaptor protein) signaling pathways between intestinal epithelial cells of young vs older animals? Are other mechanisms at play, or is it only the (acquired) resistance which differs between young vs old?
Answer: Thank the reviewer. We add one paragraph in the Discussion as following:
“Pigs at all stages can be infected by PEDV, however both morbidity and mortality of suckling piglets within 7 days of age are very high. The mortality of fattening pigs and sows after infection is very low and most of the infected pigs show recessive or transient diarrhea. Adult pigs are matured with mucosal immunity and often immunized with vaccines compared with newborn piglets. Very likely, there is a difference in the PRR-adaptor cell signaling pathways between intestinal epithelial cells of young vs older pigs. Additionally, other resistance mechanism may be also at play, which differs between young vs old pigs.”
-quantification of the plaque assays has not been performed / is not included in the manuscript. Without such quantification (plaque count), it is difficult for the reader to judge the scientific value of the observations. Have enough replicates been performed?
Answer: The results of the plaque assay were quantitatively analyzed as shown in the Fig S3 below. All the results of the plaque assays were from repeated experiments, with Figure 3C more than 3 repeated experiments and the rest 2-3 repeated experiments. The analysis results were consistent with the results of RT-qPCR and WB tests.
Minor items:
-Materials&Methods: have the monoclonal antibodies used for the adaptor proteins been verified to cross-detect porcine host-proteins?
-M&M: please include some (supplementary) data showing the N-specific mAb generated indeed exerts the target antigen-specificity and affinity as aimed for.
Answer: The PEDV N protein monoclonal antibody used in the experiment has been confirmed to be highly specific by indirect ELISA, WB and IFA. It doesn’t have cross reaction with host proteins. The results are shown in the Fig S1 as following.
-Results figure 2b: what is the explanation for the other expression bands observed in some of the transfected cell lines outside the corresponding adaptor protein? Is it reflecting cross-reactivity of the Ab’s used?
Answer: The GFP antibody used is very specific. The other expression bands beyond the corresponding major adaptor proteins are likely adaptor related proteins which are expressed in transfected cells.
-line 39: Belongs to the alpha-coronavirus family?
Answer: Yes.
-line 201: despite the fact?
Answer: No such words in the manuscript.
-Results line 211 typo: PRRSV = PEDV
Corrected.
-line 220 proteins (plural)
Corrected.
-Figure 3: positioning in of the A and B panels
The position of Fig 3A and 3B has been exchanged.
Line 314: ‘worldwide swine disease’: do not know if this is accepted phrasing / swine disease with worldwide distribution?
Corrected as following: a swine disease with worldwide distribution which has important economic significance

Reviewer 2 Report
The presented work is of a very good standard and corresponds to your focus in your journal. I only have small comments.
In the abstract section, I recommend providing brief information about the methodology of the experiment.
In the abstract section I also recommend stating that the PEDV strain used in this study was a mutant strain of PEDV AJ1102 81.
For greater clarity, I recommend providing a list of abbreviations with explanations.
Author Response
Comments and Suggestions for Authors
The presented work is of a very good standard and corresponds to your focus in your journal. I only have small comments.
Thank the reviewer for appreciation of our work.
In the abstract section, I recommend providing brief information about the methodology of the experiment.
Answer: Several methodologies have been added in the Abstract.
In the abstract section I also recommend stating that the PEDV strain used in this study was a mutant strain of PEDV AJ1102.
Answer: PEDV AJ1102 like strain has been added in the Abstract.
For greater clarity, I recommend providing a list of abbreviations with explanations.
Answer: The abbreviations with explanations have been added as much as possible in the revision.